# Fast Charging Systems for Passenger Electric Vehicles

**Rick Wolbertus \* and Robert van den Hoed \***

Department of Urban Technology, Faculty of Technology, Amsterdam University of Applied Sciences, 1097 DZ Amsterdam, The Netherlands

**\*** Correspondence: r.wolbertus@hva.nl (R.W.); robert@etransition.eu (R.v.d.H.)

**Abstract:** This paper explores current and potential future use of fast charging stations for electric passenger vehicles. The aim of the paper is to analyse current charging patterns at fast charging stations and the role of fast charging among different charging options. These patterns are explored along the lines of the technical capabilities of the vehicles and it is found that with increasing battery capacity the need for fast charging decreases. However, for those vehicles with large charging capacities there are indications that fast charging is perceived as more convenient as these are used more often. Such results indicate a larger share for fast charging if charging capacities increase in the future. Results from a spatial analysis show that most fast charging is done at a considerable distance from home, suggesting mostly 'on the road' charging sessions. Some fast charging sessions are relatively close to home, especially for those without private home charging access. This shows some future potential for fast charging in cities with many on-street parking facilities.

**Keywords:** fast charging; passenger cars; data

## 1. Introduction

Fast charging systems for electric vehicles are becoming increasingly important to facilitate the transition to zero emission mobility. Until just a few years ago, the dominant fast charging power for passenger vehicles was 50 kW, while the most recent systems enable charging at speeds of at least 175 kW or even 350 kW. In commercial and logistic fleets, even higher fast charging rates are applied (up to 1 MW)—such as for buses and heavy-duty trucks. Such systems enable much smoother long-distance travel for electric vehicles (EVs) with shorter charging times and more efficient utilization of the stations from a business perspective.

Developments in fast charging lead to particular challenges related to the integration of charging facilities in the grid, power requirements, impacts of fast charging on batteries (with or without new battery chemistries), added value of storage facilities, as well as compatibility with renewable energy sources. Insights into the use patterns of fast charging stations within daily travel patterns is, therefore, necessary. Could these patterns be changing with technological developments, and are these developments strong enough to change the current charging paradigm? Similar to level 2 (<22 kW) public charging infrastructures, the rollout and planning of fast charging facilities requires major investments, while best practices are likely to be developed where the business case improves.

Currently fast charging is mainly used as a back-up option on the road [1] when batteries run empty. However, some would consider fast charging to become dominant if charging speeds approach gasoline refuelling times [2]. However, it is still unclear if such recharging practices would meet the charging needs of EV drivers. Potential lock-in effects could arise if level 2 charging is abundant and EV drivers associate charging with parking and its potential benefits.

The motivation for this study is that so far research on the role of fast charging among all the options of fast charging is missing. Several studies have studied the split among charging modes but

research into what role fast charging plays and how the technical capabilities of the car play a major role has been missing. This paper explores the concept of fast charging and sees how it is applied by electrified passenger vehicles in the Netherlands by exploring a dataset with charging records from nearly 20,000 EV drivers in the Netherlands. It uses spatial-temporal analysis to try to explain fast charging choices. In doing so it helps us to understand the current utilisation of fast charging stations but it also allows us to forecast the role of fast charging stations in the future.

## 2. Fast Charging Systems

### 2.1. Defining Fast Charging

To start, a definition of fast charging is needed. Botsford and Szczepanek [3] defined it as anything that is not slow charging, but such a definition leaves confusion about what exactly is slow charging. Therefore, they refer to the California Air Resources Board definition of a charge that enables the vehicle to travel for 100 miles in 10 min. However, with an average consumption of 0.2 kWh/km this would require a charger of more than 180 kW, far above the average deployed power at the moment. Other papers discussing fast charging do not define fast charging but leave it up to the readers to interpret the term [1,4]. Some others [5] only refer to fast charging as charging on direct current (DC) but others argue that alternating current (AC) charging can also provide at least power up to 43 kW. Nicholas and Hall [6] refer to this AC standard but still choose to exclude it due to its limited use in practice and decide to set the bar at 36 kW provided through DC.

For this paper the definitions by the Society of Automobile Engineers (SAE) and International Electrotechnical Commission (IEC) as mentioned by Nicholas and Hall [6] are used, but the 43 kW AC power option is included. This is done so because from a consumer's perspective it is irrelevant whether or not the power is provided through AC or DC. Fast charging is thus defined as charging at any station that provides at least 43 kW of power. Note that it is defined as charging at such a station, actual charging speed might differ because charging power can drop during the charging process as the batteries' state-of-charge increases.

### 2.2. Technical Properties of Fast Charging

The actual charging speed at fast charging stations is limited by both the properties of the charger and the properties of the car, including its battery. Limitations on the charger side are usually limited to the charging cable and available power at the site. Given the high powers that are used for charging (newer standards allow up to 350 kW), a sufficient cable diameter is needed to reduce resistance and heat formation. Increasing diameter also implies increasing weight making the cable not fit for use by consumers. Latest research [7], therefore, focuses on liquid-cooled charging cables to reduce size and heat transfer. Higher capacity charging (>1 MW) for e.g., buses and trucks make use of overhead charging which removes the need for a cable.

Battery limitations also play an important role. Increased temperatures in the battery due to higher currents in the battery lead to faster battery degradation [8,9]. EV manufacturers, therefore, decrease charging speed as the battery is filled. Sufficient battery management systems can control the charging process to reduce the stress on the batteries [10,11]. Active cooling of the battery, often liquid-based, seems to be necessary to increase charging rates in the future.

### 2.3. Fast Charging as a Behaviour

Fast charging, besides its technical properties and limitations, is also part of a behavioural pattern of the EV driver. Fast charging is part of the charging options EV drivers have. Multiple researchers [12–14] have shown that private (home) charging (if available) is the most dominant charging mode. If not available nearby public level 2 charging stations are used as a substitute [15]. Together with workplace charging, these charging while parking modes are found to be dominant in the charging regime often accounting for more than 90% of all energy charged. Other modes include

level 2 charging in public or at semi-public locations (such as parking garages) for destination charging. Such charging generally takes a very small market share (<5%)

Fast charging only takes about a 5% to 10% market in the total energy charged by EVs. It is considered a back-up option if the battery runs out during longer trips or when there is insufficient public level 2 charging infrastructure in the neighbourhood [12]. Newer research has also pointed to some new use cases for fast charging, such as killing time between appointments [16], opting for free charging or convenient charging at supermarkets. Additional markets can be sought in charging for taxis [17], shared vehicles [5] or freight transport [18]. These modes require more up time and slower charging is, therefore, often not considered a solution.

Motivations for the choice for fast charging instead of other modes are a gap in literature. To our knowledge fast charging choices have only been studied in Japan [19] and in the United States [20]. While the Japanese study focused on complete trips and the US study only on price changes, both noted a significant effect of the price on charging. The Japanese study also found that EV drivers do not prefer to detour from their route for their charging, but when doing so prefer locations with facilities such shops at gas stations.

### 2.4. The Role of Fast Charging for Passenger Vehicles in the Future

Although fast charging stations currently have a big role to fulfil long-distance recharging needs, possible growth in charging speeds might shift their potential. Major cities in China, such as Beijing already deployed as much fast charging stations as level 2 charging stations [21]. This is an attempt to fulfil the needs of urban EV drivers that do not own a private parking spot and thus a designated charger. Although the limited charging speeds mean that such an option is not preferred by the EV driver at the moment, it could be that such preferences change. Most cars can only charge with charging speeds up to 100 kW, which results in charging times of over 30 min to fill the car up to 80%. Such charging times have translated into preferences for charging while parking. With an increase in battery capacity and better battery management systems it can be foreseen than EVs will allow higher charging speeds. If charging times drop significantly, this could result in a switch in preferences for EV drivers towards fast charging. Stated choice experiments have at least shown that shorter charging times increase the attractiveness of fast chargers over regular level 2 charging stations [16]. So far analysis of charging preferences depending on charging speed have not been made.

### 3. Methodology

#### 3.1. Data Collection

Data were retrieved from MultiTankCard, a large mobility service provider in the Netherlands. The records contained any charging session in which the charging card is used to gain access to the given charging station. Note that in some situations access to the charging station is not regulated through the charging card in some cases of private home charging, free charging or use of dedicated charging networks such as the Tesla Supercharger network. For home charging, the majority of EV drivers log their charging sessions as many of the EV drivers' employers refund any mobility costs.

Records of the charging data are recorded per person, which are identified through a passenger scrambled authentication ID number. To ensure the privacy of the EV driver, no passenger records were collected and address information was limited to partial postal codes. In total, charging records were provided for 17,191 EV drivers who recorded 1,004,111 charging sessions in 2019. Data contain information about timing information about the charging session, the volume, location and charging station type. Each charging station is defined with an unique Charge Point ID. If the charging stations has multiple connectors the connector used is given with *X. Table 1 gives an overview of the recorded data per charging session and an exemplar of each variable.

**Table 1.** Data description.

| Variable | Exemplar |
|---|---|
| Start time | 1 March 2019 12:30 |
| End Time | 1 March 2019 17:00 |
| Volume (kWh) | 4.867 |
| ZIP Code (partial) | 3478 |
| City | Zaandam |
| Country | The Netherlands |
| Charge Point Type | 4 |
| Authentication ID | 34THR0959760 |
| Charge Point ID | NLLMSE0178897*2 |

The charge point type category is based on the power the charging station can provide (Table 2). This gives information about fast charging stations (value 41, 5 or 50). To enrich the information about charging station type the dataset is combined with a set on public charging stations [15]. Workplace charging stations are those that are not public, home or fast charging but have a median start time of between 6:00 a.m. and 9:00 a.m. Not classified charging stations are classified as semi-public/else.

**Table 2.** Data description of type of charging stations available in the dataset.

| Charge_Point_Type | |
|---|---|
| Value | Description |
| 0 | UNSPECIFIED |
| 1 | AC < 3.7 kW (1 phase) |
| 2 | AC 3.7 kW (1 phase) |
| 21 | AC 7.4 kW (1 phase) |
| 3 | AC 11 kW |
| 4 | AC 22 kW |
| 41 | AC ≥ 43 kW |
| 5 | DC ≥ 50 kW |
| 50 | DC 20−50 kW |
| TL | Home Charging |

### 3.2. Data Transformation and Analysis

Data are filtered for impossible and improbable session, which results in leaving out sessions with a volume above 100 kWh (more than current battery capacity in battery electric passenger vehicles) and charging sessions with a volume below 0.01 kWh. Filters are also applied to the duration of the charging sessions to exclude charging sessions shorter than 1 min and longer than 720 h. Additionally, this paper focuses only on EV drivers who recorded at least 30 charging sessions in The Netherlands. Data of drivers with fewer charging sessions do not contain enough information about actual charging patterns. It was observed that these drivers also substantially charged less energy, not representative of actual drivers.

Additional information about each of the EV drivers is derived from the charging data. For each of the drivers a private location is estimated based on the charging pattern. This is either the private home charge point, or the most often used (semi-)public charging station given that it matches a home-charging pattern. To establish if the public charging station is used for on-street home charging, the median start time should be between 15:00 and 22:00. The battery size of the EV driver is obtained from the maximum value of the volume variable. It is assumed, given that at least 30 charging sessions are observed, that EV drivers charge at least once with a nearly empty battery. Battery size is grouped being either below 16 kWh (plug-in hybrid EVs) or per 20 kWh battery capacity.

Data on the location and distances between the charging stations were limited to a partial postal code. For these partial postal codes, shapefiles were obtained and using the rgeos package in R

we obtained the centroid coordinates. Driving distances between these coordinates were obtained using the osrm package in R. Analysis of differences between groups in this paper were undertaken using chi-square and *t*-test analysis. Additional insight in the distribution of data are visualized using bar charts and boxplots.

## 4. Results

### 4.1. Descriptive Statistics

At fast charging stations, the average charging session is 15.4 kWh (Figure 1). The distribution is left centred with the majority of charging sessions close to the average but a long tail of some charging sessions with a large amount of energy charged (up to 85 kWh). With an approximation of 0.15−0.2 km/kWh, this implies an average range added of about 77–102 km. These findings are in line with other research [1,16,22]. Average connection time at the charging station is 23 min ($\sigma$ = 14 min). This implies an average speed of 41 kW during the charging session. This does not mean that the whole charging session was at this charging speed as this also depends on other factors such as the state-of-charge and the battery management system of the vehicle. The maximum speed observed over the entire charging session was 148 kW. Therefore, it is likely that the dataset also contains chargers that can deliver 150 or 175 kW.

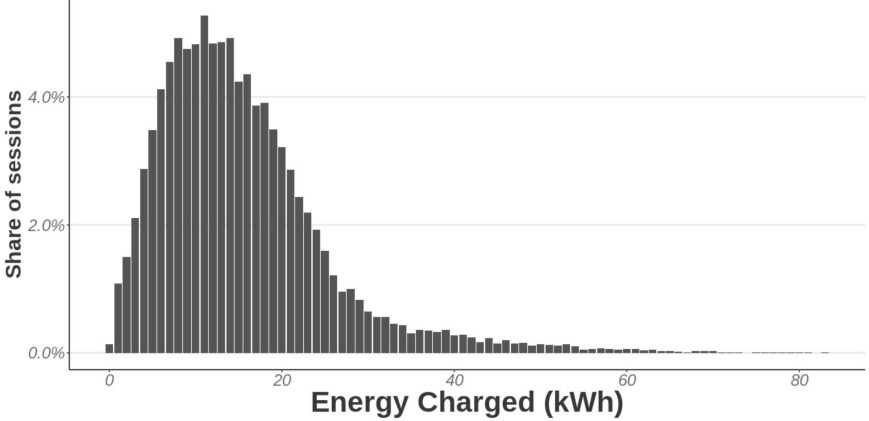

**Figure 1.** Distribution of energy charged at fast charging stations.

Given the distribution of charging sessions over the day, Figure 2 shows that the majority of charging sessions occur during the afternoon. A peak can be observed between 11:00 and 15:00. This is line with other observations at fast charging stations but completely differs from patterns at home [23] or public level 2 charging stations which have clear peaks in the morning and evening. This hints at the fact that fast charging serves a different need for the EV driver compared to level 2 charging.

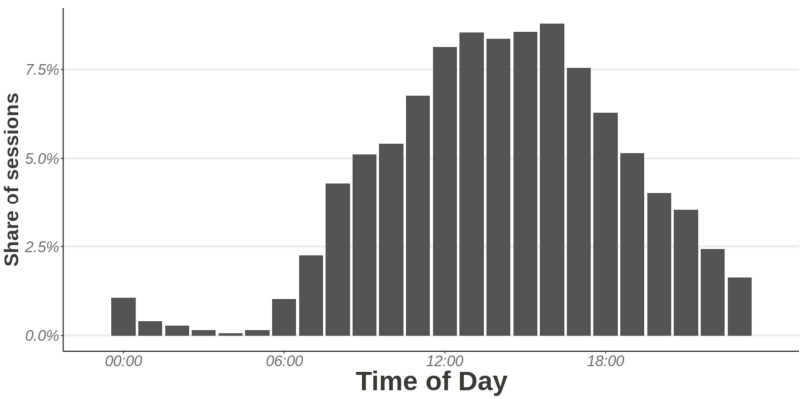

**Figure 2.** Distribution of fast charging sessions across the day.

This suggestion can confirm the results of the data analysis in Figure 3. These findings are in line with earlier research about these charging sessions. Fast charging plays a small role across the day and slightly peaks in the afternoon. However, its share never reaches far above 10% while there are clear dominant modes in the morning (workplace) and afternoon (home).

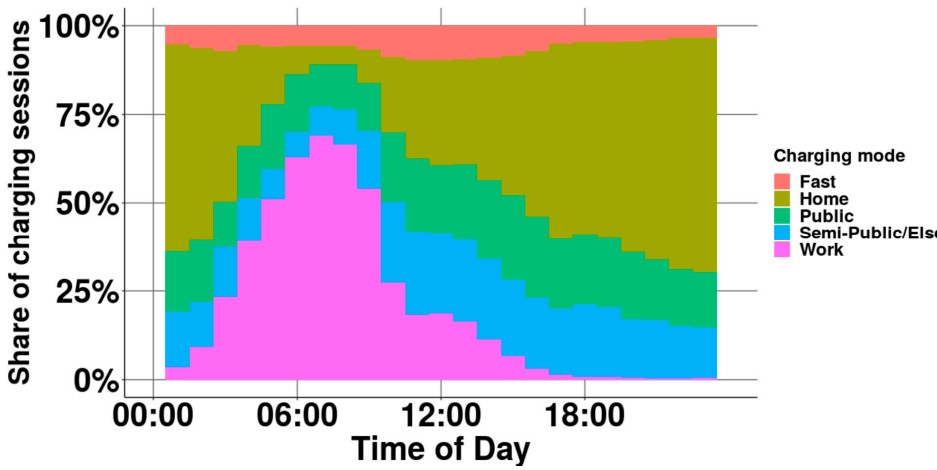

**Figure 3.** Share of charging sessions across charging during the day.

### 4.2. Fast Charging as Part of Charging Habits

Fast charging stations are only one option from which the EV driver makes use to fulfil its charging needs. Most used options include home charging, workplace charging and (semi-)public level 2 charging. On average the EV driver only charges 22 kWh/month at a fast charging station, approximately 1.5 charging session. The large standard deviations show there are large differences between EV drivers in fast charging station use. The average fast charged energy is only a small share of the total EV drivers' charging habit (Table 3). The share for vehicles with a smaller battery capacity is more than double those with a large battery pack although the absolute numbers are nearly equal. This shows that the driving of an EV is a limiting factor in the total distance travelled.

**Table 3.** Descriptive statistics of energy charged per month at fast charging stations per range of battery capacity.

| Battery Capacity (kWh) | Mean (kWh/Month) | Min (kWh/Month) | Max (kWh/Month) | Standard Deviation (kWh/Month) | Mean kWh/Month Across All Types | Share of Energy (%) at Fast Charging |
|---|---|---|---|---|---|---|
| 16–30 | 21.6 | 0 | 395.0 | 36.5 | 165.1 | 13.0% |
| 30–50 | 21.7 | 0 | 919.8 | 45.4 | 276.7 | 7.9% |
| 50–70 | 23.0 | 0 | 603.7 | 45.3 | 335.3 | 6.9% |
| 70–100 | 23.2 | 0 | 366.8 | 39.3 | 363.8 | 6.3% |

This also results in EVs with a smaller battery (16−30 kWh) having to fill a larger part of their battery at fast charging stations (Figure 4). These drivers on average nearly fill half of their battery at these stations. For the larger battery packs (70–100 kWh) this drops to below 30%, with the distribution showing that a majority of the sessions even have a smaller part of their battery filled during charging.

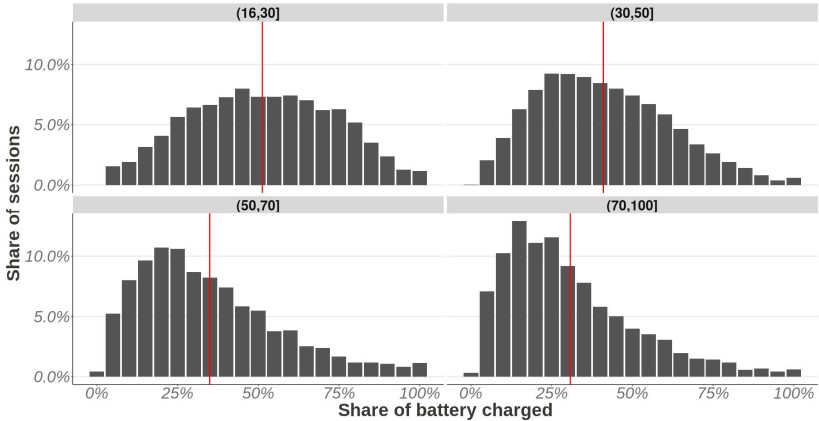

**Figure 4.** Energy charged relative to battery capacity for ranges of battery capacity. Red line indicates average value.

In line with the energy charged, fast charging is not the major share of charging sessions for EV drivers. On average, only 3% of charging sessions is as fast charging stations. Figure 5 illustrates how this differs across battery capacity and across groups who do or do not have access to private charging at home. Most notably it can be observed that full electric vehicles with a small battery (16−30 kWh) are in most need of fast charging. These differences between the battery size groups are statistically significant $X^2$ (16, N = 34138) $p < 0.01$. Also, those with private charging access need less fast charging (only 4% compared to 11%), $t$ (10367) = 28.314, $p < 0.01$. It is hypothesized that a fear of not having access to public charging near home leads to more fast charging sessions to ensure a follow-up trip.

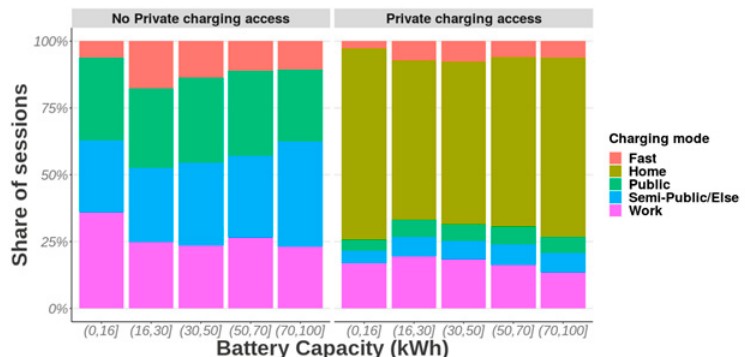

**Figure 5.** Distribution of charging sessions across different type of charging stations. Data are displayed for different battery size capacities and the availability of private charging.

### 4.3. Fast Charging as a Behaviour

So far the most common hypothesis is that the use of fast charging stations is mainly for topping up to be able to complete the journey or reach the next charging station [24,25]. Some newer research also points to other use cases for fast charging such as having time left anyway [16] or the influence of free charging [19]. Other use cases than passenger vehicles such as taxis [17] also have different behavioural patterns. These other behavioural patterns could include fast charging as a back-up for regular for public home charging in cities, but research on these patterns is missing. To study such relationships the distance from home when using a fast charging station is interesting as it can reveal back-up for public home charging, especially in relationship with the battery capacity of the car, but also give insight into how fast charging stations are used differently in long-distance travel.

Results of this analysis in Figure 6 show that among full electric vehicles with smaller batteries especially there is more need for fast charging close to home. As shown earlier in Figure 4, this is often due to a lack of private charging availability. On average the distance between the fast charging 67 km,

but the distribution is rather wide. About 26% of the fast charging sessions take place within 20 km of the most used charging location.

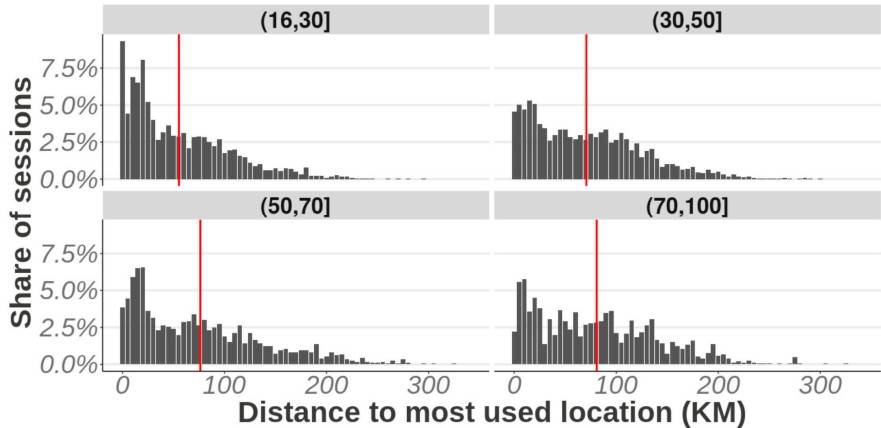

**Figure 6.** Distribution of distances to most used location for fast charging sessions for given range of battery capacity. Red line indicates average value.

It is possible that the short distances could be due to nearly not making the home location to recharge which would imply a relationship between the distance from the most used location and the energy charged. Figure 7 shows a scatter graph of the energy charged in relation to distance from the most used location. The blue line is a model estimation of the relationship between the two. A simple linear model finds a positive (coefficient = 0.0157) and significant ($p < 0.01$) correlation between distance and the energy charged. However, the variance explained by the model is rather small ($R^2 = 0.0077$), showing that distance from home or work only has a small predictive value on the energy charged. It was also hypothesized that the relation between energy and distance would flatten above a certain distance as one cannot charge more than the maximum battery capacity, regardless how far from home one is. Figure 7, however, does not show such a relationship.

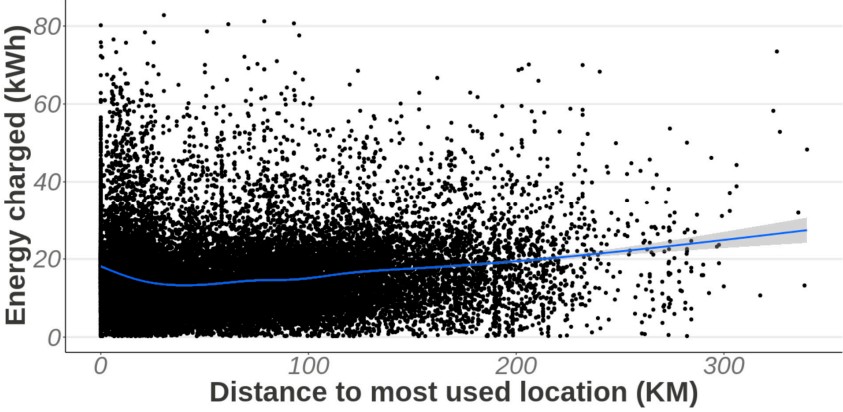

**Figure 7.** Scatterplot with regression line (blue) of energy charged at fast charging stations in relation to distance to most used charging location.

The evidence for the need for close-to-home or inner-city fast charging is mixed. A significant amount of the charging sessions take place close to home, but this is still at locations next to the highway (>90%). Such results should, however, also be interpreted in the light that the majority of charging stations is at highway locations. It is estimated that at least 75% of the fast charging locations in the dataset are along highways, implying that highway locations are currently relatively better used compared to inner-city locations.

The final analysis takes a look how the fast charging capacity, measured as maximum per driver of the average charging speed across the entire charging session, influences the average charging volume per charging session and the total charging volume observed across the year. As seen in Table 4, charging is a necessity for those with smaller battery capacities. On average they charge less per session, but quite substantially more accumulated across the year. Remarkable is that those with higher charging powers and higher battery capacities the total charging volume increases. It is unclear whether this is due to higher total charging needs or due to perceived convenience of the fast charging process.

**Table 4.** Charging volume depending on charging and battery capacity.

| Charging Capacity (kW) | Battery Capacity (kWh) | Number of EV Drivers | Mean Volume/Session(kWh) | Volume/Year/EV Driver (kWh) |
|---|---|---|---|---|
| (0,50] | (16,30] | 551 | 12.74 | 103.60 |
| (0,50] | (30,50] | 1471 | 13.48 | 93.68 |
| (0,50] | (50,70] | 711 | 17.91 | 65.03 |
| (0,50] | (70,100] | 322 | 18.53 | 79.20 |
| (50,100] | (16,30] | 37 | 11.79 | 24.22 |
| (50,100] | (30,50] | 62 | 17.92 | 23.99 |
| (50,100] | (50,70] | 34 | 31.46 | 50.90 |
| (50,100] | (70,100] | 21 | 32.61 | 51.24 |
| (100,150] | (16,30] | 311 | 13.79 | 66.26 |
| (100,150] | (30,50] | 639 | 15.82 | 46.13 |
| (100,150] | (50,70] | 423 | 23.53 | 70.60 |
| (100,150] | (70,100] | 222 | 29.07 | 121.15 |

## 5. Conclusions

This research has highlighted the most important aspects of fast charging for the use of passenger electric vehicles. An overview of the literature and the data analysis presented has shown that fast charging mainly plays a role in long-distance travel, or at least in case when the battery runs empty on the road. This paper has explored the potential of fast charging in the future given technological developments and the charging behaviour of the EV driver.

Data have shown that fast charging has distinctly different patterns in terms of energy delivered and the timing of the charging session compared to other charging modes. The analysis suggests a relatively small role for fast charging across the day and is especially relevant for those drivers who own a full electric vehicle with a smaller battery pack (<30 kWh). These drivers tend to drive and therefore charge less than those drivers with larger battery packs, meaning that their absolute contribution to the energy charged at fast charging stations is rather similar.

Given the developments in battery technology [26] and the shift towards more long-range EVs [27] it can be questioned whether the role of fast charging continues to grow in the future. There are indications that for those with large battery packs and higher charging capacities there is a convenience in fast charging which could continue to grow with further technological improvements. However, such observations are only preliminary and should also be interpreted in the light that those with higher battery also tend to drive more in generally. The turning point for more charging convenience could be at higher charging speeds than 150 kW as this would really limit the charging speed. Better availability of fast charging stations could then be considered more attractive instead of cruising for level 2 charging stations in the city. Further research could look into the motivations of EV drivers for choosing fast charging and which factors play a crucial role. Such research could confirm the possibility that at faster charging speeds EV drivers are more inclined to use fast charging stations at a regular basis.

An analysis of the distances of fast charging sessions to the home locations shows that most charging is done 'on the road'. There is some room for close to home 'fast charging' in the future, although currently this is mostly done by vehicles with a shorter range, which could indicate that fast

charging station operators could potentially expand to inner-city locations. Demand for fast charging could also be induced simply by its mere presence. The data that were analysed were mostly from highway locations because fast charging stations can be found there.

Generally, fast charging stations have a mixed future perspective. Given the trend in battery capacities in vehicles there is a decreasing demand for 'necessity' charging sessions. The convenience of higher charging speeds might counter this decreasing need in necessity charging. This would open up pathways to fast charging at new locations such as in the city. The path towards this will be crucial, and currently the slower charging modes are dominant, both in the number of charging stations and sessions. Fast charging stations will have to cover significant ground in order to become obtain a significant share in the charging mix.

**Author Contributions:** R.W. has done the data analysis and wrote the manuscript in consultation with R.v.d.H. All authors have read and agreed to the published version of the manuscript.

**Funding:** This work is part of the Future Charging project (RAAK.PRO03.128) funded by Stichting Innovatie Alliantie (SIA).

**Conflicts of Interest:** Data has been provided by MultiTankcard, a large mobility service provider in the Netherlands at no cost or any obligations.

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
