# Peer review of "Fast Charging Systems for Passenger Electric Vehicles"

_wevj, doi:10.3390/wevj11040073_

Round 1

Reviewer 1 Report

This is an interesting work but some aspects must be improved before publishing:

_ Some parts of the paper in which the results are analyzed are confusing, and the conclusions drawn seem self-contradictory. It is essential to better analyze the results, proposing the possible explanations and evaluating why some of the hypothetical conclusions are more feasible than others. Authors are encouraged to improve the following sections: Abstract, 4.3. Fast charging as a behaviour, and 5. Conclusion.
In the future research, surveys to drivers of electric vehicles should be carried out in order to know the origin of charging habits. In this way, some of the hypothetical conclusions in the paper could be confirmed.

_ It is necessary to improve the quality of the presentation. For example, the format of the text and the figures changes along the paper.

Reviewer 2 Report

I have had the pleasure of reviewing your manuscript to which I share the same field. I find spatiotemporal charging behavioural researches like these to be highly attractive and yearn for more works in this space. To this end I am suggesting that this manuscript be published with minor amendments according to my comments listed in the following:

  1. Minor linguistic and typographical errors are present. It would be beneficial for this manuscript to be proofread prior to publication. e.g. "fast in the mix (L7), "20.000" (thousand separator is a comma in English) (L40), "Us study" (L91). 
  2. Inconsistent formatting. Font changes in L20, Tables of different font sizes and broken headers. Also I assume the notation e.g. (16,30] means 16-30 though I find this notation strange.
  3. Please expand the last paragraph of the Introduction to include the main motivation and elaborate on the contributions that this manuscript presents.
  4. L76. How is fast charging part of a behavioural pattern of the EV driver? When it's stated "Fast charging is part of the charging options EV drivers have," I assume home/workplace charging and level 2 charging are the other options. Though when referred to Figure 3 does it indicate that "Prive" (possibly a typo of private) indicates home charging? If so can we please get some convention consistencies in this manuscript?
  5. The relevancy between Section 2.4 and the results and discussion presented in this manuscript is unclear.
  6. Table 2. How is "value" used in the dataset? They do not seem to be arbitrary numbers. 
  7. L137. It might also be a good idea to apply time filters in addition to energy filters to discount charging events that are too long/too short.
  8. L206. How is it determined that users do not have access to private home charging? Was there a survey conducted among EV owners?
  9. Figure 7 and Table 4 are not cited in text. Also what does the blue line indicate in Figure 7?
  10. What is the motivation behind the analysis for the distances of fast charging stations relative to the user's home location?

Round 2

Reviewer 1 Report

Accept in present form

Reviewer 2 Report

I have reviewed the manuscript and found the revisions to be satisfactory for publication.